



# EXPLORATORY EXPERIMENTS ON PRE-ACTIVATED FREEZING NUCLEATION ON MERCURIC IODIDE

Gabor Vali

Department of Atmospheric Science, University of Wyoming, Laramie, WY, USA.

**Correspondence:** vali@uwyo.edu

**Abstract.** Pre-activation of freezing nucleation was examined in laboratory experiments with mercuric iodide suspensions in water. The experiments followed the procedure designed by Edwards, Evans and Zipper (1970) but employed multiple sample drops and many repetitions of the pre-activation cycle. The results obtained confirm the basic findings of the earlier work and refine it. By also drawing on the results of Seeley and Seidler (2001), pre-activated freezing nucleation (PFN in this work) is

analyzed in search of constraints that help define the process responsible for it. No firm conclusions are reached, but evidence is accumulated pointing to the role of definite structures being involved in PFN, similar to the role of sites in heterogeneous freezing nucleation in general. PFN differs from pore condensation and freezing described by Marcolli (2020) and David et al. (2020) in that it takes place in liquid water. Further exploration of this process can help understading ice nucleation at the basic level and in its practical manifestations. The results call attention to an ice nucleation pathway hitherto barely explored

and which can be expected to have consequences in how ice nucleation occurs in atmospheric clouds and in other systems.

## 1 Introduction

The phenomenon that is to be examined here consists of observations of freezing nucleation by suspended particles just a few degrees below the melting point that follows prior freezing of the sample and heating to just a few degrees above the melting point. This specific cycle was first reported by Edwards, Evans and Zipper (1970; EEZ70 in the following). Many

other manifestations of nucleation depending on the prior history of the sample are known.

    In this paper, the phenomenon of freezing at a temperature slightly below the melting point following prior freezing and subsequent moderate heating above the melting point is referred to as "pre-activated freezing nucleation", PFN[1]. The phenomenon has also been referred to in the literature as enhancement or as memory effect. PFN is perhaps the most neutral expression to use, in that it does not imply a specific process. "Enhancement" seems to imply that a given INP or nucleating site is responsible

for nucleation both in the normal mode and in previously exposed cases. In contrast, "memory effect" puts emphasis on the fact that prior freezing is a precondition for the observed high freezing temperatures. To retain some flexibility in the description and as a reminder that different interpretations of the results are possible, while PFN will be used most frequently, other terms will also be employed occasionally to describe the phenomenon.

---

[1]A clearer definition of PFN will be given in the following section and its meaning refined in Section 5.2





Because ice nucleation is so inaccessible to direct observation, concepts of the process rely on empirical evidence bounding

the conditions for it. PFN is mostly of interest from the point of view of broadening or restricting those concepts. PFN provides additional evidence to be incorporated into the known set of constraints about how to view the process of ice nucleation. More specifically, this paper focuses only on freezing nucleation, as emphasized by the term PFN. Much recent work focused on the pore condensation and freezing process (PCF) which takes place in the vapor phase.

## 2   Previous results

EEZ70 reported experiments with mercuric iodide, $HgI_2$, and other substances. $HgI_2$ is a moderately effective ice nucleating substance. Observed activity is comparable to what was seen with some minerals in similar experiments: nucleation temperatures of $-8°C$ and lower. However, $HgI_2$ is one of a very few substances for which pre-activation is known to be possible.

In the EEZ70 experiments, a single drop of water containing some $HgI_2$ was suspended in a pressure cell and surrounded by an inert fluid. The cell was then subjected to a prescribed sequence of temperatures. The principal finding of EEZ70 is that ice

nucleation at some $T_f$ followed by continued cooling to $T < T_C$, and further followed by warming to a temperature $T_w < T_D$ leads to nucleation on subsequent cooling at $T_f^*$ barely below the melting point of ice. If warmed to above $T_D$, subsequent nucleation takes place at $T_f << T_f^*$. In addition to $HgI_2$, EEZ70 report on similar experiments with other substances, at different pressures and with salt solutions. The results are interpreted, following the results of Evans (1967), in terms of the formation of a two-dimensional ice-like monolayer on the substrate which facilitates nucleation of bulk ice unless destroyed

by heating above $T_D$. Prior cooling below $T_C$ is necessary for the monolayer to change from disordered to ordered form. Fig. 1 is a schematic representation of this process.

The basic features of the pre-activation described above were shown by Seeley and Seidler (2001a; abbreviated as SS01) to be also exhibited with aliphatic alcohols as ice nucleators. Their experiments were performed at 1 bar using a single drop coated with the nucleating Langmuir film of the aliphatic alcohol and placed on a cooling stage. Many hundreds of cycles

of cooling and heating were performed varying the warm limit $T_w$ in stepwise fashion. Three different alcohols were used giving different $T_D$ values but all leading to similar values of $T_f^*$ between $-6$ and $-10°C$. In SS01, there is some gradual lowering of freezing temperature as $T_w$ is raised and an abrupt shift to lower values at $T_D$; there is no further lowering of the freezing temperatures beyond that. No results are given regarding $T_C$; apparently cooling below a given limit was not a necessary condition for pre-activation in their experiments. An important aspect of the results in SS01 is that for any given $T_w$,

the freezing temperatures vary over a range of approximately $4°C$ in random fashion over the many cycles of the experiments. They showed the same variations of nucleation temperatures for the aliphatic alcohols without pre-activation. In all cases, the frequency of freezing as a function of temperature, $R(T)$ in their notation, is interpreted in terms of CNT.

SS01 considered their results to be in accord with the monolayer explanation of EEZ70. A stronger dependence of $R(T)$ on the exponential factor of the CNT equation is claimed to be consistent with a monolayer being responsible for the pre-

activation and not any "dimensional" change such as a rare defect in the Langmuir layer. This, in effect, reduces emphasis on the nucleating substrate, or nucleating sites on it, and focuses attention on changes in the water structure near the substrate.





SS01 point to the findings of Majewski et al. (1994) for evidence on the ordering of a monolayer on the aliphatic alcohols and suggest experiments to examine how the changes in the monolayer relate to the dependence of pre-activation on $T_D$.

The findings of EEZ70 and of SS01 are about pre-activation of freezing nucleation. Pre-activation was shown to also exist

for deposition nucleation by Fournier d'Albe (1949), Mason and Maybank (1958), Higuchi and Fukuta (1966) and Roberts and Hallett (1968). In addition to the potential for an ice layer to be retained on the surface, the possibility that liquid water or ice in cavities, pores or crevices of a substrate can exist outside the normal boundaries of phase changes for the bulk phases has also been proposed for explaining pre-activation. More recently, pore condensation and freezing received strong empirical evidence and theoretical support (Marcolli, 2014, 2020; Wagner et al. 2016; David et al., 2019, 2020); this process, if coupled with

nucleation sites within the pore, lead to ice formation at low supersaturations. The process has not been linked to pre-activation in the liquid.

The current work confirms the main findings of EEZ70, and has some parallels with the results of SS01. A detailed comparison with these works will be given in Section 5 after the presentation of the experimental method for this work in Section 3 and the data obtained in Section 4. The experiments were performed in the early 1970s with much simpler equipment than now

available . Yet, the paucity of information on PFN gives relevance to the data then obtained. A preliminary summary of this work was given in Vali (1992).

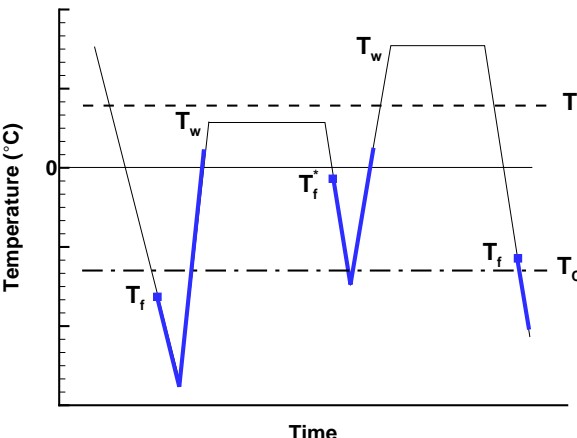

**Figure 1.** Schematic representation of the findings of EEZ70 and defining the notation used in this paper. The blue lines represent ice and the blue squares points of nucleation.



## 3 Experimental technique

### 3.1 Sample preparation

All experiments here described were performed with the same batch of the mercuric iodide, $HgI_2$, supplied by Mollinckrodt
Chemicals (St. Louis, MI, USA; Lot WYLS, 99% purity). It is described as a soft material. Mercuric iodide is only slightly
soluble in water ($6x10^{-5}$ g/mL); the solute effect on the depression of the melting point is ignored in this work. Weighed
amounts of the red $HgI_2$ powder, as received, were added to 100 mL of distilled water to reach concentrations of 0.02 and 0.04
g mL$^{-1}$. Supernatant was drawn into a sterile syringe for dispensing drops of 0.01 cm$^{-3}$ volume onto the cold stage. In most
experiments, 121 drops were tested simultaneously. Of these, 22 were used as controls with purified water.

Mercuric iodide is listed as being light-sensitive. The suspended particles were illuminated during the freezing tests. This
may account for some of the variations that were observed over the durations of the experiments. No attempt was made to
quantitate this.

The goal of these experiments was to examine the memory effect with no emphasis on characterizing the activity of $HgI_2$
per se. This meant that no special effort was made to achieve close control of the sample preparation. No rigid protocol was
set, partly because rigid control of the process would have been unachievable with the available means. As a result, no data
are available on particle size distributions in the drops. Variations in sample preparation and handling, and the spread of the
experiments over a period of a year, led to variations in the activity observed in the samples. However, the data of interest
here derived from repeated cycles of freezing with given sets of drops and those observations are independent of the sample to
sample variations. Additionally, even a given set of drops cannot be taken to be perfectly identical, due to possible settling of
particles and the need to refill the syringe various times for the production of a set of drops for a run.

In spite of the practical difficulties referred to above, the number of particles of $HgI_2$ was undoubtedly high enough to make
variations relatively unimportant. Assuming a mean particle diameter of $0.1\mu$m, the number of particles per drop was on the
order of $10^{11}$, an ample number to consider each drop to have the same chance to contain an INP. The experiments detected
the INP with the highest freezing temperature in a drop, making that event unique.

### 95 3.2 Freezing experiments

The experiments were carried out in the same manner as those described by Vali (2008). They were performed intermittently
over a period of about a year (1972-73) when the drop-freezing apparatus (Vali, 1971 ) was available. Briefly, the apparatus
consisted of a cold stage of a 1 cm thick copper block of 10x10 cm dimensions. The block was covered with aluminum foil
using a heat-conducting cream to reduce temperature variations. A thin silicone varnish was applied to provide a hydrophobic
surface. Cooling of the cold stage was via Peltier elements and a circulating liquid heat exchanger, controlled to be at $-1°$C
min$^{-1}$. Temperature measurement was obtained with a calibrated thermocouple. A digital temperature display and the drop
array were photographed at intervals of 15 s. Illumination was optimized to have the cold stage nearly completely dark, thus
enabling freezing to be detected by reflection by the ice within the drops. After all drops were frozen the stage was heated at
roughly 1°C min$^{-1}$ to a maximum preset value of $T_w$. Temperature overshoot at $T_w$ was held to <0.2°C.





The photographic records were evaluated by experienced technicians. The images were projected on a table one frame at a time, stepping from frame to frame on a manual command (push button) by the technician after a thorough visual scan on the image. It was possible to move backward and forward to compare adjacent images. When a change in opacity was discerned for a drop, the temperature reading from that frame was written over the image of the drop. These records were subsequently entered into spreadsheets for computer analysis. Detection of freezing from the change in drop opacity was rather critical in

these experiments because many of the freezing events were at just a few degrees Celsius below the 0°C. The processing of the photographic film (16 mm) was tailored to achieve good but not excessive contrast and to be reproducible. To control for the unavoidable uncertainty in the detection of freezing events due to human subjectivity, some of the readings of the film records were repeated by two individuals. Differences were $< 0.5°$C for the majority of cases at few degrees below 0°C and less than that at lower temperatures. In all, freezing events were reliably detected at $T_f < -2°$C and colder and for this work that is

viewed as the detection limit.

Measurement errors of the stage temperature, as well as non-uniformities across the stage were smaller than the uncertainty resulting from determination of the moment of freezing. Therefore, the overall accuracy of data here reported is taken as 0.5°C, but occasional larger errors can't be ruled out.

### 3.3   Types of experiments

In order to explore different aspects of the memory effect, experiments of various types were performed, falling into two major groups. One group was aimed at determining the limits of heating above 0°C that still produces some enhanced nucleation. In this group are the following experiments: (i) Gradual increase of the warm limit, $T_w$ from one run to the next, (ii) Gradual decrease of the warm limit, (iii) Alternating high and low warm limits. The other group of experiments was performed to examine the reproducibility of the memory effect and the influence of time: (iv) Varying lengths of time at the warm limit, and

(v) Repeats of the same warm limit.

In all experiments, the initial run was performed right after placing the drops on the cold stage. These runs started at room temperature which was not controlled but was 20±2°C. Cooling was continued until all drops were frozen.

Warming of the drops to only a few degrees above 0°C, and only over a short time, a valid concern arises about the possibility of some ice being retained in the drops. All evidence points to this not having been the case. In general terms, some supercooling

was required for all drops to freeze as indicated by sudden changes in opacity at the moment of nucleation. Gradual freezing that started at 0°C would have led to gradual darkening of the drop images. Results to be presented in a later section with varying length of time above 0°C provides further proof for the absence of bulk ice when repeat freezing cycles are started.

The number of sample drops varied from experiment to experiment because in most cases two or more different dilutions of the suspension were tested simultaneously. Results are reported only for the dilutions exhibiting clear PFN. One or two rows

of drops of distilled water were also included for control.





## 4  Results

### 4.1  Nucleus spectra

The $HgI_2$ suspensions used in these experiments (after initial tests to arrive at a particle concentration in the working range of the experiments) exhibited moderate activity. The cumulative concentration of INPs, defined by Vali (1971 and in Eq. 9 of
Vali et al. (2014), is shown in Fig. 2 for the various experiments to be described in this paper. Data here displayed are from the initial run of each experimental series when the drop array was first cooled from room temperatures.

There is considerable variation among the experiments due to variations in the degree of dispersion of the $HgI_2$ powder in water and the degree of settling that took place before the supernatant was withdrawn for producing the sample drops. This variation was not of particular concern for this work and no special effort was made to reduce the variability. The main concern
was to have a spread of freezing temperatures in the range -5°C to -20°C.

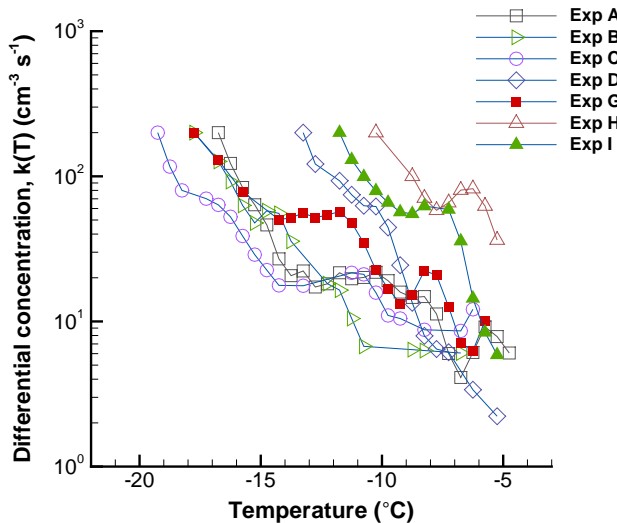

**Figure 2.** Cumulative nucleus spectra, $K(T)$ for the $HgI_2$ suspensions used in the various experiments.

### 4.2  Determining the "warm limit" above 0°C for PFN

Three series of experiments were performed to define the upper limit of temperature which allows PFN to occur. In two experiments the warm limit $T_w$ was increased gradually from one run to the next with the same set of drops. In one series the warm limit was gradually decreased. In all experiments the stage was held at $T_w$ for 5 minutes between runs.
In experiment A, after the initial run, ten more runs were performed with values of $T_w$ = 1.5, 2.0, 2.5, 3.0, 3.5, 4.0, 4.5, 5.0, 6.0 and 10°C. In Exp. B a shorter series was performed with $T_w$ = 10.0, 1.5, 2.0, 3.0, 4.0, and 5.0°C. A series of runs with



**Table 1.** Average freezing temperatures, $\overline{T_f}$, observed in the three series of experiments with gradually increasing and gradually decreasing warm limits, $T_w$.

| | Run | 0 | 1 | 2 | 3 | 4 | 5 | 6 | 7 | 8 | 9 | 10 |
|---|---|---|---|---|---|---|---|---|---|---|---|---|
| Exp. A | $T_w$ | | 1.5 | 2.0 | 2.5 | 3.0 | 3.5 | 4.0 | 4.5 | 5.0 | 6.0 | 10.0 |
| | $\overline{T_f}$ | -10.8 | -3.1 | -3.4 | -3.4 | -3.7 | -4.1 | -8.6 | -8.9 | -10.5 | -12.8 | -13.8 |
| | Run | 0 | 1 | 2 | 3 | 4 | 5 | 6 | | | | |
| Exp. B | $T_w$ | | 10.0 | 1.5 | 2.0 | 3.0 | 4.0 | 5.0 | | | | |
| | $\overline{T_f}$ | -13.7 | -13.1 | -4.2 | -3.8 | -4.3 | -12.7 | -12.9 | | | | |
| | Run | 0 | 1 | 2 | 3 | 4 | 5 | 6 | | | | |
| Exp. C | $T_w$ | | 10. | 8.0 | 8.0 | 5.0 | 4.0 | 3.0 | | | | |
| | $\overline{T_f}$ | -13.0 | -13.3 | -14.5 | -14.1 | -14.5 | -13.6 | -5.5 | | | | |

gradually decreasing warm limits was executed in Experiment C, with $T_w$ = 10.0, 8.0, 8.0, 5.0, 4.0 and 3.0°C.The number of drops tested simultaneously was 77 for Exp. A and 33 for Exp. B and Exp. C.

A summary of the results for these series of runs in given in Table 1 and in Figures 3, 4 and 5. All three series indicate a pronounced change between $T_w$ = 3°C and $T_w$ = 4°C independently of the direction of the change from increasing or decreasing warm limits. This is the main finding, bracketing the value of $T_D$. The detection limit for freezing of the sample drops is another factor limiting a fully clear delineation of the magnitude of the PFN effect observed. When taking into account the range of freezing temperatures indicated by the vertical bars in the figures it may be noted that there is some overlap between the events on either side of the major jump.

Another way to illustrate the impact of raising the warm limit past $T_D$ is to look at the fraction of drops showing significantly elevated freezing temperatures. This is shown in Table 2. The cutoff values used to generate the table varied somewhat in order to have adequate sample sizes. A large jump near $T_w = +3.5$°C in the percentage of drops freezing above the cutoff is evident in all three experiments. However, it is important to note that some PFN can be seen even at $T_w = +5$°C and $T_w = +6$°C. That these are not artifacts is reinforced by the 0 values for the initial run and for runs with $T_w = +10$°C.

The description given above in terms of ensemble parameters can be put into better perspective by elaborating on the variations encountered when examining individual sample drops. To this end, the temperature histories of individual drops are displayed for Exp. A in Fig.6. Apart from the steady $T_f$ values in runs 2-4 where the detection limit restricts variations, a number of different patterns can be distinguished. For example, drops 2, 35, 56 and many others retained $T_f \approx -4$°C even to run 8 that followed $T_w = -5$°C. On the other hand, drops 1, 21, 39, 80 and others exhibit sharp decreases in $T_f$ after run 6 following $T_w = -3.5$°C. There are other patterns as well: gradual shifts, sudden changes up or down, or no systematic changes. These variations complicate the definition of $T_D$ and have to be born in mind when discussing the significance of that as a threshold value.





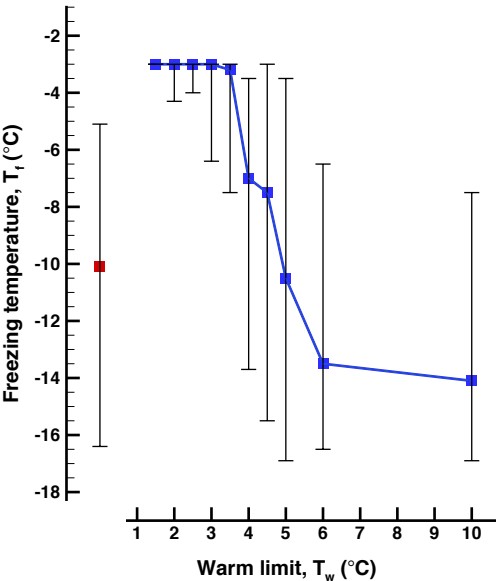

**Figure 3.** Freezing temperatures in the sequence of runs (Exp. A) with increasing warm limit, $T_w$. The first bar on the left is for the initial run started at room temperature. The 50 percentile is indicated by the square symbols and the vertical lines show the 5 and 95 percentile values.

**Table 2.** Fractions of drops with significant PFN at different warm limits. experiments A and B had the warm limit increased in successive runs, while in experiment C the warm limit was decreased from high to low values.

|  | 10.0 | 1.5 | 2.0 | 2.5 | 3.0 | 3.5 | 4.0 | 4.5 | 5.0 | 6.0 | 8.0 | 10 |
|---|---|---|---|---|---|---|---|---|---|---|---|---|
| Exp A 77 drops; -4.0 cutoff | 0 | 100.0 | 94.8 | 96.1 | 85.7 | 83.1 | 20.8 | 24.7 | 7.8 | 7 0 |  | 0 |
| Exp B 33 drops; -5.5 cutoff | 0 | 97 | 97 |  | 94 |  | 3 |  | 9 |  |  |  |
| Exp C 33 drops; -5.5 cutoff |  |  |  |  | 61 |  | 0 |  | 0 | 0 | 0 | 0 |

## 4.3 The effect of time duration samples are held at the warm limit

Data in the preceding section was produced with a 5-min holding period at the warm limit. Clearly it was of interest to test how
shorter of longer exposures to temperatures above the melting point would effect the degree of PFN. This factor was tested, although only in one experiment (Exp. D), with the time at $+1.5°C$ altered between 1 and 5 minutes. Two sequences were tested with an overnight gap between them.

In a sense the 1-min holding time was also a test of whether incomplete melting may have led to subsequent freezing right at $0°C$. Because of that, extra care was taken in the data reduction in this experiment to detect freezing of the drops as early





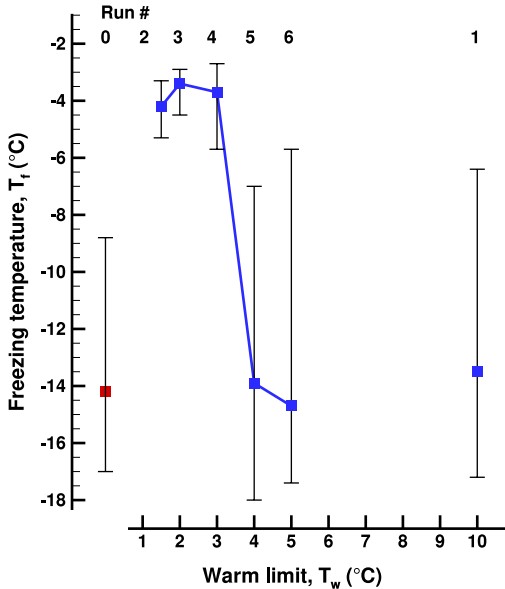

**Figure 4.** Freezing temperatures in a sequence of runs (Exp. B) with increasing warm limit, $T_w$, after run 1 with $T_w = +10°$C. Run 0 was started at room temperature. The 50% value is indicated by the square symbols and the vertical lines show the 5 and 95 percentile values.

as possible. Events were recorded starting at $-2°$C. No difference was found in that threshold between the 1 and 5 minute holding times. Results for the sequence of runs are shown in Fig. 7. The two repetitions are shown as one series.

Mean temperatures for the 6 runs were: -10.2, -2.3, -2.8, -11.1, -3.3, -4.4°C. The large differences between runs 0 and 3 starting from room temperature and runs 1 and 4 after $T_w = +1.5°$C is clear showing a strong PFN effect. The decrease seen between runs with 1 or 5 min at the warm limit is $0.5°$C for the first pair and $1.1°$C for the second pair. These differences are

statistically significant to better than $0.01\%$ level. The evidence is for time exposure at the warm limit to be important, with the degree of memory effect slightly reduced with longer exposure. In hindsight, the experiments described in the preceding section could have been sharpened by using 1 minute hold times at the warm limit, but the 5 minute period seemed safer for avoiding that bulk ice remain in the drops.

A better appreciation of the change with time exposure can be gained from looking at the magnitudes of the changes drop

by drop as shown in Fig. 8. Most of the changes are small in the first pair of runs (1 to 2) but are larger for the second set (4 to 5). There are positive as well as negative changes, the positive ones indicating a higher freezing temperature after 5 minutes than after 1 minute at the warm limit. The 90% ranges were 0.6 to $-2.5°$C in the first pair and $+1.45$ to $-6.0°$C for the second pair. This type of scatter in the way $T_f$ changes for individual drops is seen in all runs and will be examined further in the next section.





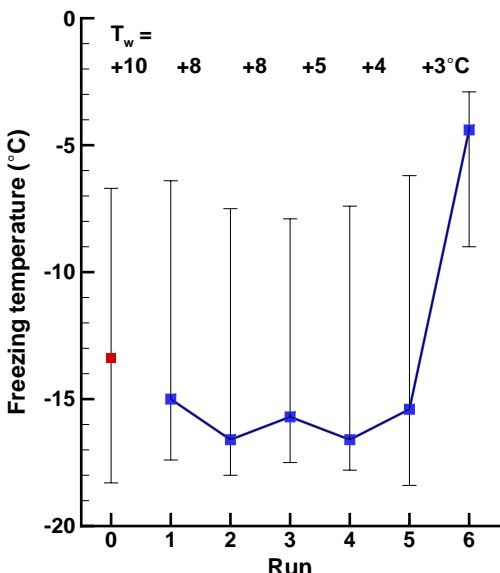

**Figure 5.** Freezing temperatures in a sequence of runs (Exp. C) with decreasing warm limit, $T_w$. The 50% is indicated by the square symbols and the vertical lines show the 10 and 90 percentile values.

### 4.4 Variability and repeatability of nucleation temperatures with $HgI_2$

It has already been noted that there is considerable variation among drops in how they respond to the various sequences of PFN tests. Since that variability is a potential indication of what the underlying process is for the PFN phenomenon, some of the manifestations of that variability are illustrated in the following, with focus on the sequences of $T_f^*$ for individual drops. Each drop was certain to contain a large number of particles of $HgI_2$. Assuming $0.1\,\mu m$ particle size, for the mass concentrations used in these experiments the number of particles per drop is of the order $10^{11}$. This should assure good statistical equivalence. Even so, there are considerable differences in freezing temperatures among drops. This makes it clear that the nucleation events in each drop are linked to rare particles or parts thereof.

In all of the following, repeatability refers to multiple nucleation events of a drop within a narrow temperature range. That range is $\approx 1\,to\,2°C$, which is meaningful only when the range of freezing temperatures for the full sample set of drops is considerably larger than that.

#### 4.4.1 Run to run correlations

In the current experiments, for runs with strong PFN and hence a small range of $T_f^*$ values, correlations can not be meaningfully evaluated due to the detection limit and since the range of freezing temperatures in these runs is only about twice the precision

(c) Author(s) 2020. CC BY 4.0 License.





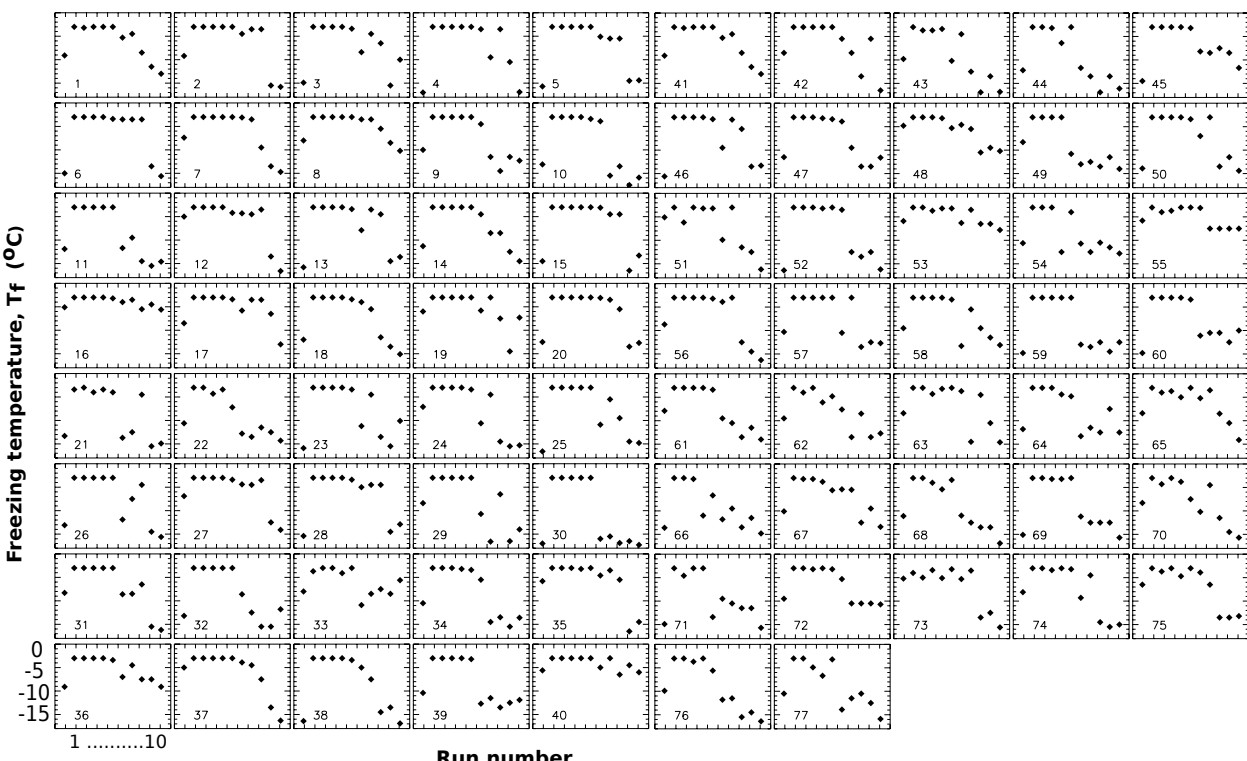

**Figure 6.** Freezing temperatures in Exp. A for run 0 and runs 1 to 10 for individual drops. Run 0 is the initial run, 1 to 10 are runs with controlled $T_w$ values as shown in Table 1.

**Table 3.** Correlation coefficients for selected pairs of runs. Values with an asterisk are for runs involving PFN.

| Runs | 0-1 | 1-2 | 2-3 | 3-4 | 4-5 | 5-6 | 6-7 | 7-8 | 8-9 | 9-10 |
|---|---|---|---|---|---|---|---|---|---|---|
| Exp A | | | | | | | 0.45* | 0.45* | 0.18* | 0.52 |
| Exp B | 0.49 | | | | | 0.59 | | | | |
| Exp C | 0.74 | 0.85 | 0.93 | 0.92 | 0.70 | | | | | |
| Exp I | 0.39 | 0.48 | 0.65 | 0.58 | 0.66 | | 0.62* | 0.69* | 0.61* | 0.44* |

of the temperature data. Only for runs with higher warm limits or with reasonably large spreads in freezing temperatures is

this analysis possible. A summary of the correlation coefficients is given for selected pairs of runs in Table 3. The correlation coefficients are generally low. Highest values are for Exp. C, probably because of the relatively flat spectrum for this experiment (cf. Fig. 2).

In addition to the correlations, individual drop histories are a revealing way to examine the variability, or repeatability, of freezing temperatures. The "spiderweb" diagram in Fig. 9 shows, in a different way than Fig. 6, the freezing temperatures for





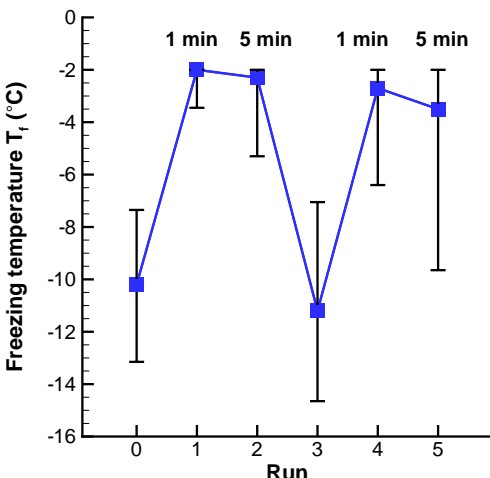

**Figure 7.** Freezing temperatures in a sequence of runs with 1 and 5 minute holding times at $T_w = +1.5°$C. The 50 percentile is indicated by symbols and the vertical lines show the 5 and 95 percentile values.

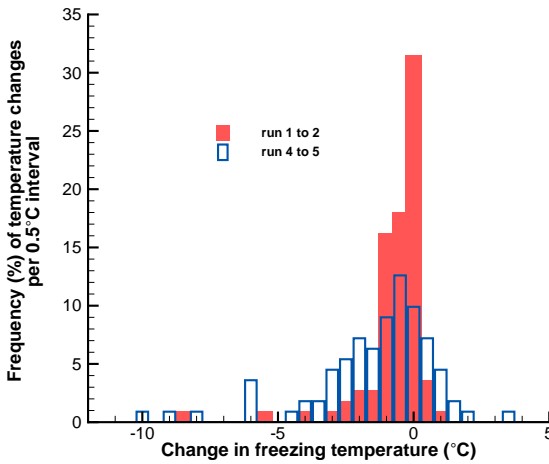

**Figure 8.** Histograms of the changes in the freezing temperatures of drops between runs with 1 and 5 minutes at $T_w = +1.5°$C.

individual drops in Exp. A. The spread of $T_f$ values in the initial run is represented in the spectra shown in Fig. 2. The rise in $T_f^*$ after $T_w = +1.5°$C brings all drops to a narrow range. That range starts to spread, with smaller and larger excursions, in subsequent runs. Beyond $T_w = +3°$C run-to-run changes of large magnitudes are seen both in the positive and negative





directions. In the region between $T_w = +3.5$ and $+6°$C, some drops retain freezing temperatures near the highest values while others drop to lower values. Some return to high $T_f$ in subsequent runs.

To illustrate the degree of variation in the changes in freezing temperatures, frequency distributions of these changes are shown in Fig. 10 for the four runs where the relaxation of PFN begins. Changes in mean temperatures for these four cases were -0.42, -0.27, -3.9 and -0.13°C. Gradually increasing spread is seen as well as a shift to negative values for the pair $T_w = +3.5$ to $+4.0°$C. Changes remain centered on 0 for the last pair $T_w = +4.0$ to $+4.5°$C but with changes of up to $\pm10°$C in magnitude. Changes to lower temperatures are more expected that those toward higher ones.

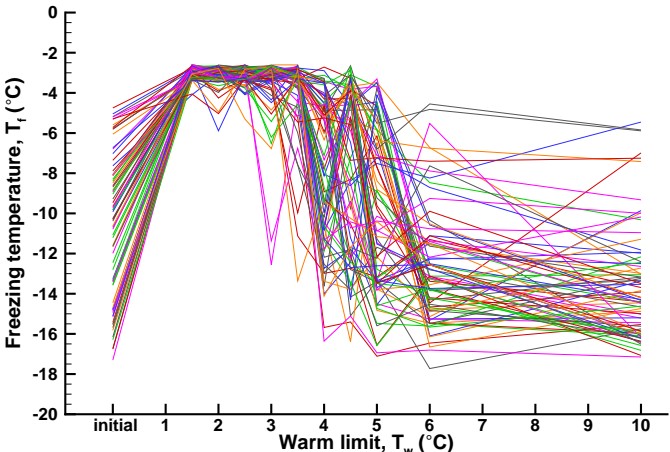

**Figure 9.** "Spiderweb" diagram of individual drop histories for the sequence of runs with increasing warm limits in Exp. A. A statistical representation of these obervations is given in Fig. 3.

Spiderweb diagrams are shown in Fig. 11 and Fig. 12 for Exp. B and Exp. C. The overall pattern of high PFN is evident for $T_w \le +3°$C in both cases. So is also the large variability in freezing temperatures from run to run for given drops. An extreme example is seen in Fig. 11 (Exp. B) with one drop falling to a low $T_f$ for $T_w = +2$ and $+3°$C. There is a similar, smaller magnitude event for two drops at $T_w = +3°$C in Fig/ 9 but with the high PFN regained after a low freezing temperature. The evidence points to the possibility that PFN may be regained after a new cycle of freezing and heating to above $0°$C even after 230   a previous cycle in which there was less, or no enhancement.

**4.4.2   Sequence of runs with high and low values of the warm limit**

Perhaps the best data set for examining the repeatability of the nucleation events is the series of runs (Exp. I) in which a warm limit well above the transition value, $T_{w,1...5} = +10°$C was repeated for 5 runs, followed by 5 runs with $T_{w,6...10} = +1.5°$C. The observed ranges of freezing temperatures are shown in Fig. 13 in the format used earlier for other experiments.

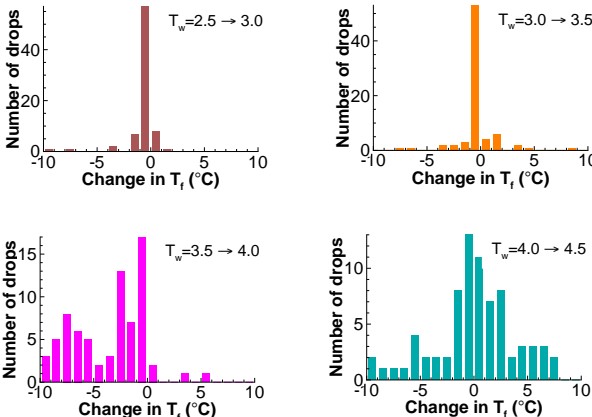

**Figure 10.** Histograms of the changes in freezing temperatures between successive pairs of runs in Exp. A

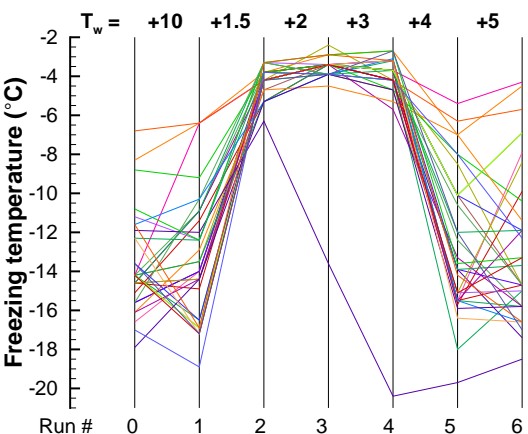

**Figure 11.** "Spiderweb" diagram of individual drop histories for the sequence of runs with increasing warm limits in Exp. B. A statistical representation of these observations is given in Fig. 4.

The difference between the two halves of the experiment stands out quite clearly. The difference for the two different warm limits is $\overline{T^*_{f,6...10}} - \overline{T_{f,1...5}} = (-4.6) - (-8.9) = 4.3°C$. This increase with PFN is less than the corresponding change of $10.5°C$ in Exp. A (Section 4.2), probably because the initial activity was already higher than in that experiment (cf. Fig. 2).

The gradual decreases in the mean freezing temperature for the first 5 runs is $-0.3°C$, and $-0.08°C$ for $T_w = +1.5°C$. The decreases for individual drops have considerable spreads: the 10 to 90-percentile values for the run-to-run changes are -2.3 and 240    $+1.7°C$ for the runs with $T_w = +10°C$, and -2.0 and +1.9 for the runs following $T_w = +1.5°C$.



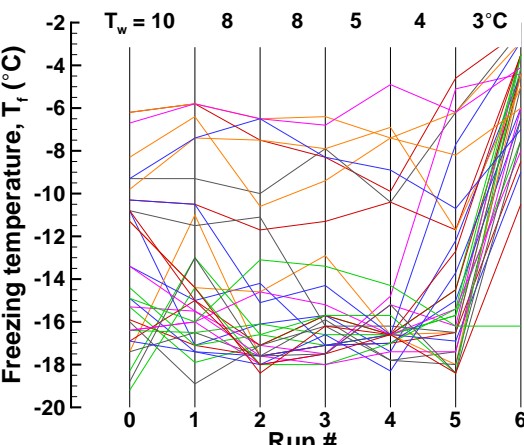

**Figure 12.** "Spiderweb" diagram of individual drop histories for the sequence of runs with decreasing warm limits in Exp. C. A statistical representation of these obervations is given in Fig. 5.

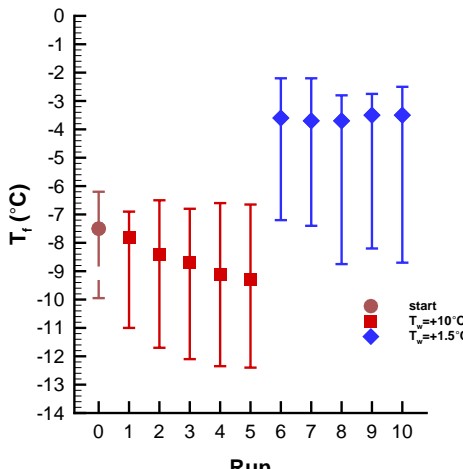

**Figure 13.** Range of freezing temperatures in Exp. I consisting of a sequence of runs with high and low warm limits. The 50 percentile is indicated by symbols and the vertical lines show the 10 and 90 percentile values.

Data from this experiment lends itself well to examining the correlation between pairs of runs because there is a reasonably large spread of $T_f$ for the $T_w = +1.5°C$ runs. Correlation coefficients for successive pairs of the these runs are 0.62, 0.69, 0.61





and 0.44. The values for $T_w = +10°C$ are 0.48, 0.65, 0.58 and 0.66. These values are comparable to those observed in Exp. A and B but are lower than those seen in Exp. C (cf. Table 3.

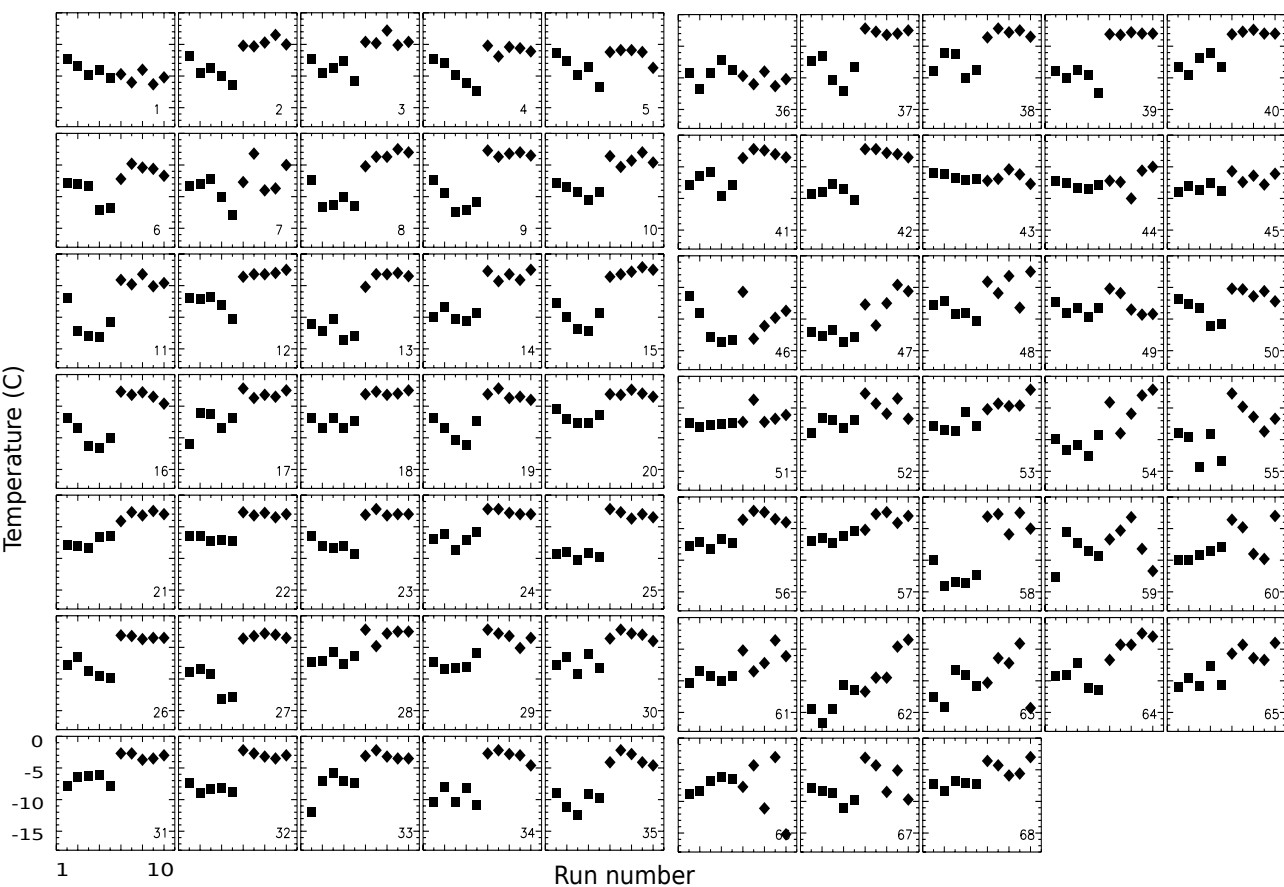

**Figure 14.** Freezing temperatures of 68 drops in Exp. I. The first 5 points (square symbols) in each panel are for runs 1...5 with $T_w = +10°C$ and the last 5 points (diamond symbols) for runs 6...10 with $T_w = +1.5°C$.

A full appreciation of the variety of sequences of the freezing temperatures of different drops is best gained from a display of the sequences of freezing temperatures for each drop in Fig. 14. Inspection of the figure reveals that some drops show very little discernible PFN. Examples of this are drops 1, 36, 43, and 66. On the other hand, 36 out of the 68 drops have $\overline{T_f} > -4°C$ for $T_w = +1.5°C$ while there are none for the runs with the higher $T_w$.

A considerable variety of patterns can be discerned in the sequences. Many sequences, or parts of sequences contain jumps 250 in irregular fashion, but in many more some pattern can be seen. Monotonic increases and decreases exist over three or four runs, perhaps with gaps. Some sequences appear to be arcs. These characteristics were seen in all the experiments. The lack of a uniform response for all drops poses a major quandary for explaining the results. Quantitating these patterns and distinguishing between random variations and perceived patterns are evidently not easy and, due to the small number of runs, one can expect





that the strength of any test be limited. Nonetheless, these signatures are the only available indicators of the stability, or lack

of, of nucleating sites. The following analyses are directed to this aim.

The meaning of run-to-run correlations (cf. Table 3) with $r$ values near 0.6 for this experiment is somewhat ambiguous. A stronger signal emerges in comparison with randomized freezing temperatures. Randomization tests were performed by re-ordering the observed freezing temperatures of drops according to a sequence created by placing the random numbers in an icreasing order. For Exp I. the $T_f$ values were scrambled for runs 2, 3, 4, 5 from the first group and for runs 7,8,9, and 10

from the second group (cf. Fig. 13), for comparisons with runs 1 and 6 respectively. The correlation coefficients using these scrambled runs dropped to between -0.2 and +0.1 for both the +10 and +1.5°C runs, showing essentially no match between $T_f$ values from a run and values from the randomized set. On this basis, the $r = 0.6$ and similar values acquire some significance as indications of repeatability. This seems to hold for freezing with or without PFN.

Another characterization of the individual drop sequences was made by fitting linear equations to the 5 points corresponding

to each of the two warm limits. The means of the slopes of the these lines are -0.33 and -0.13 for the two groups but with considerable scatter about the mean; standard deviations are 0.64 and 0.65 for the two groups. As a measure of the degree of scatter in $T_f$, the mean of the absolute deviations ("$absdev$" for short) from the fitted lines was determined for each warm limit and each drop. The mean values of these parameters are $\overline{absdev_{10}} = 0.74°C$ and $\overline{absdev_{1.5}} = 0.68°C$. The values for $T_w = +1.5°C$ have clear temperature dependence ranging from 0.35 for drops with $\overline{T_f} > -4°C$ to 1.4 for drops with $\overline{T_f} < -6°C$.

The means and standard deviations of $absdev$ for the two groups are 0.74/0.39 and 0.68/0.61. After randomization these values became 1.13/0.56 and 1.14/0.6. The increased values of $absdev$ reinforce the indication of non-random patterns in $T_f$.

### 4.4.3 Alternately high and low warm limits

The previous section provided some evidence that repeated cycles at $T_w = +1.5°C$ maintain a degree of similarity in freezing temperatures. Also, some drops showed a return to high $T_f^*$ after a much lower one in an intermediate run (cf. Section 4.2). To

further examine this, two experiments (Exp. G and H) were carried out with alternate cycles of $T_w = +1.5°C$ and $T_w = +10°C$. The observed ranges of freezing temperatures for four repetitions of the cycle are shown in Fig. 15 and Fig. 16. For unknown reasons, the data from both of these experiment (Exp. G and H) yielded unexpectedly small differences between runs with high and low warm limits. In addition, Exp. H has unusually high freezing temperatures.

With all four pairs of +10 and +1.5 runs in Exp. G, the average difference is only $1.4°C$; the largest difference is for the

second pair at $2.0°C$. These are much smaller signatures of PFN than seen in other experiments. For individual drops, the 90% range of changes is -5.5 to +9.7°C. On the average 62% of the drops showed positive changes. Changes in $T_f^*$ between one run and the next with the same $T_w = +1.5°C$ are spread over a range of $\pm 8°C$. This large variability is quite different from the comparable value of $\pm 2°C$ for Exp. I (Section 4.4.2).

Nonetheless, even this series has some examples of patterns of repetition for individual drops as shown for 9 cases (a tenth

of the total) in Fig. 17. The consistent pattern for each of these suggests that these are not merely random occurrences. There are indications for freezing temperatures to repeat or change in systematic ways when either the high or low warm limits are





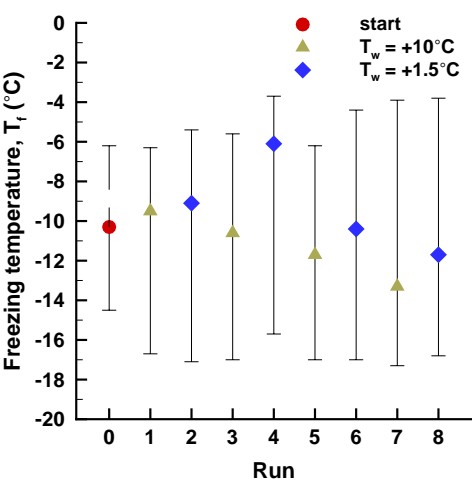

**Figure 15.** Freezing temperatures in a sequence of runs (Exp. G) with alternating high and low warm limits, $T_w$. The 50 percentile is indicated by the symbols and the vertical lines show the 5 and 95 percentile values.

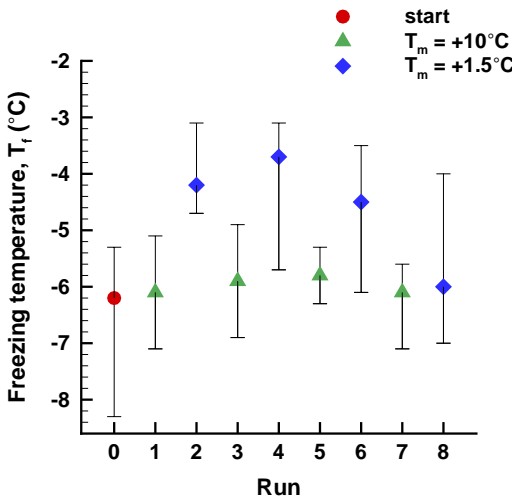

**Figure 16.** Freezing temperatures in a sequence of runs (Exp. H) with alternating high and low warm limits, $T_w$. The 50 percentile is indicated by the symbols and the vertical lines show the 5 and 95 percentile values.

applied prior to cooling. It is noteworthy that the patterns of changes for the two sets of $T_w$ appear to be independent of one another.





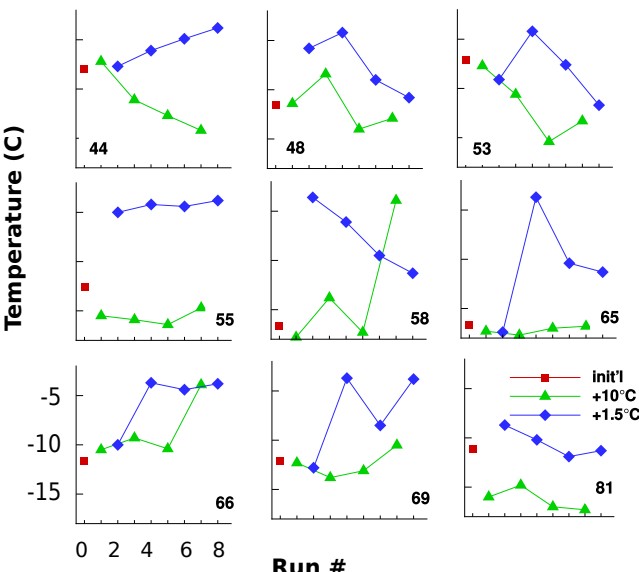

**Figure 17.** Sequences of freezing temperatures for 9 selected drops in Exp. G.

The other series of alternate warm limit experiments (carried out 10 days later) is Exp. H for which the range of freezing temperatures is shown in Fig. 16. Even the initial run had unusually high freezing temperatures and so did the runs after all $T_w = +10°C$ runs. The increases due to pre-activation are small but clearly evident for the first three cycles. The difference is statistically meaningful to better than 1% significance level.

Due to the small spread of freezing temperatures, run-to-run changes in the freezing temperatures of individual drops is also small (10 to 90% range is only $1.3°C$). Patterns of repeatability cannot meaningfully be distinguished from random variations. Re-randomization tests confirmed this.

### 4.4.4 Other tests of repeatability

For Exp. A (gradually increasing warm limits), a test was developed that provided a measure of how subsequent $T_f$ values for drops with $T_f^* > -4°C$ in a given run compared to those with with $T_f^* \leq -4°C$. The results are shown in Table 4. It is seen that, drops frozen with $T_f^* > -4°C$ in any of the runs had higher freezing temperatures in subsequent runs. This holds for all but one of the pairs of runs. Examples of the more pronounced differences are shown in bold font in the table. With this test having sufficiently large numbers of drops in each category, the test results are good indications of real differences. The level of significance of the differences was not evaluated because the differences are quite emphatic and point in the same direction as the other tests of repeatability. The pattern holds even as $T_w$ is raised. Furthermore, the difference persists at least for two subsequent runs.





**Table 4.** Average freezing temperatures $\overline{T_f^*}$ in runs $i+1$, $i+2$ $i+3$ for $n_f$ drops with $T_f^* > -4°C$ in run $i$ and the same for the $n_{uf}$ drops that froze below that limit in run i. The comparison is given for four different values of $T_w$. Bold font indicates differences exceeding $2°C$.

| $T_w$ in run i | | $i+1$ | $i+2$ | $i+3$ |
|---|---|---|---|---|
| $T_w$=+3.0 | $n_f$=66 | -3.8 | **-7.4** | **-7.6** |
| | $n_{uf}$=11 | -3.8 | **-9.6** | **-9.7** |
| $T_w$=+3.5 | $n_f$=62 | **-7.2** | **-7.3** | **-9.6** |
| | $n_{uf}$=15 | **-9.9** | **-10.4** | **-11.6** |
| $T_w$=+4.0 | $n_f$=14 | -7.3 | **-8.4** | -12.8 |
| | $n_{uf}$=63 | -8.0 | **-10.3** | -12.5 |
| $T_w$=+4.5 | $n_f$=19 | **-7.7** | -11.6 | -12.6 |
| | $n_{uf}$=58 | **-10.7** | -12.9 | -13.9 |

As mentioned in Section 4.4.3, there appears to be no clear relationship between the nucleation temperature with or without pre-activation. This finding is further examined for runs 5 to 6 in Exp. I (Section 4.4.2) and for averages of the two groups of runs in the same experiment. For both of these, the correlation coefficient is 0.3 and the corresponding scattergrams reveal that a few outliers have a strong effect. Random mixing of the values of the second of the pairs of runs leads to smaller $r$ values. It takes about 50 re-randomizations of the second run to get one of them to yield an $r$ value near the observed one. These tests

confirm the lack of correspondence between $T_f$ and $T_f^*$ for any given drop.

The same point can be made by looking at the magnitude of the $(T_f - T_f^*)$ difference drop by drop. While the mean value of the change from $T_w = +10°C$ to $T_w = +1.5°C$ is $4.6°C$ the 90% range for individual drops is $(\overline{T_f})_{+10} - (\overline{T_f^*})_{+1.5} = -0.4 \rightarrow$ $7.3°C$. The change from the last $+10°C$ run to the first $+1.5°C$ run ranges from $0.1$ to $9.3°C$. In this case, the large variation isn't entirely due to the bunching of freezing temperatures at the lower warm limit, as seen in Fig. 13 with the vertical bars.

**5   Discussion and conclusions**

To reiterate the title of the paper it is emphasized that the data presented is exploratory, not firm. Two reasons contribute to this. One is the lack of knowledge of the amount of $HgI_2$ remaining in suspension, and no information of the size distribution of the particles in the sample drops. The other is the surprisingly complex results obtained. It may be that the latter is a consequence of the former, but the evidence points to that not being the main reason. As the experiments themselves, the discussion to follow

is focused on two aspects - defining basic characteristics of the PFN phenomenon and providing some constraints for potential explanations. With this in mind, the experimental evidence is first compared with the results of EEZ70, then the results obtained with the multiplicity of samples and diverse experiments are summarized. Finally, the implications of the results are explored in view of various theories.





### 5.1 INP derived from $HgI_2$

In the current experiments the $HgI_2$ suspensions were tested in identical fashion to other materials in previous work (e.g. Vali, 2008), thus the initial runs in each series is directly comparable to what has been found for other INPs. Similarly to other materials, the activity of INPs exhibits a spread over a range of temperatures with the number of INPs increasing with decreasing temperature. This is expressed quantitatively with the nucleus spectra shown in Fig. 2. The INP concentration of 20 or 40 $g\,L^{-1}$ was selected to yield freezing events at temperatures $> -20°C$. The spectra exhibit some minor peaks but

are not consistent for all the experiments. The general trend of the spectra, as expressed by the slope of the logarithm of the concentration is in the range $\omega = 0.3...0.5$. This is comparable with the range of values in Fig. 2(b) and Table 1 of Vali (2014). From this perspective there is nothing unusual regarding the INP activity of $HgI_2$. Somewhat unexpectedly, the magnitude of the variation in INP concentration seen in this graph exceeds what would result from the factor 2 variation in the amount of $HgI_2$ added. It is unknown to what extent this was due to alterations of the stored powder or variations in sample preparation.

Light sensitivity, different degrees of clumping and different degrees of settling in the water are potential factors. Since the samples proved to be stable once dispersed as drops and the main focus was on the sequences of runs not comparisons among runs, the sample variability was not important. The only exception to stability of INP activity in the sample drops (apart from those associated with PFN) is the gradual loss over the first 6 run in Fig. 13.

It is likely that the variability of $HgI_2$ as INP is related in some way to the relatively low run-to-run correlations (Sec-

tion 4.4.1) and to drop to drop variations in PFN. No hints were found in this work as to what the underlying cause might be.

### 5.2 Limiting conditions for PFN.

The first point that can be made is that the our findings are close to those of EEZ70[2] regarding the upper temperature limit for PFN to survive. EEZ70 gave $T_D = +3°C$. Data here presented show that the limit is not sharp. As described in in Section 4.2,

the major loss of PFN is near $T_D \approx +3.5°C$ but some PFN can be found even after heating to $T_D \approx +5°C$. That is not a wide range of the warm limit but enough to indicate that the nucleation events leading to PFN are not unlike nucleation at supercooled temperatures, i.e. dependent on specific configurations or sizes either of the substrate or of bound molecular clusters.

The fraction of sample drops exhibiting PFN as the warm limit is increased is given in Table 2. The transition from 100% to 10% for Exp. A extends from $T - w = +2$ to near $+5°C$. An important distinction regarding what to consider the limit $T_D$ is

that the range of variation is not just due to drop to drop differences which might be associated with the variations in the size or number of $HgI_2$ particles in different drops. Some fluctuations are also exhibited by single drops, i.e. with given INP contents. This is quite clearly seen in the many lines crossing back and forth in the transition region in Figs. 9, 11, and 12 and in Fig. 6.

The second point demonstrated by the data is that the duration of exposure to $T - w$ has an impact. Longer times reduce the degree of PFN observed. This point can't be compared with the EEZ70 results as no time scale was specified in their paper.

---

[2]EEZ70 contains results with $HgI_2$ and various other substances. In this work only $HgI_2$ was tested.





The third point to emphasize concerns the dependence of PFN on the degree of prior cooling necessary. EEZ70 quotes a minimum of $T_C = -20°C$ for PFN to occur. The experiments did not specifically test this assertion, but some contradiction can be noted in that the lowest temperature reached in some of tests was considerably higher. In Exp. D (Fig. 7), Exp. I (Fig. 13) and Exp. H (Fig. 16) cooling proceeded to only about $-16°C$ and PFN were nonetheless observed.

        Finally, as mentioned already, the minimum supercooling for freezing after pre-activation was not well defined in this work,
because the detection limit for freezing events was near $-2°C$. However, the data presented here show that neither does PFN always occur close to 0°C as stated in EEZ70. Data here presented show that there is a large range of temperatures over which PFN is exhibited, namely freezing events occurring at temperatures above what is observed when the sample is cooled initially or after warming to $T_w > +8°C$. Taking the temperature at which <5% of the samples were frozen in the initial run and 95% froze after activation to define the limit for PFN, the values obtained are $-5°C$ in Exp. A (Fig. 3), near $-8°C$ for Exp B.
(Fig. 4) and near $-7°C$ for Exp. I (Fig. 13). It is clear from these figures that as $T_w$ increases greater degree of supercooling is required for subsequent $T_f^*$. Based on this, it is fair to consider $T_f^* \approx -6°C$ as a lower limit for PFN on $HgI_2$ INPs in these experiments. That limit is not due to a limitation of the effectiveness of PFN, though that is also a factor, but to the presence of nucleating sites effective at that value even without pre-activation.

        Based on the foregoing, the approximate boundaries for PFN to occur are sketched in Fig. 18. The experimental constraint
are also indicated. The shading is an attempt to represent the likelihood of freezing in different zones. The blurred boundaries of the shaded area are indicative of the variability in the data shown in the bar diagrams of Figs. 3, 4, and 5. Also, the separation between what is PFN and what is due to unaffected INPs is somewhat arbitrary as described in the preceding paragraph. The value of $T_f^* \approx -6°C$ serves as an indicator of the limit. The main purpose of Fig. 18 is to emphasize the strong evidence for PFN in the upper left-hand corner of the shaded area, and that PFN extend from there to considerable ranges both in the warm
limit and in the resulting freezing temperatures. The effect of time at $T_w$ is not represented in Fig. 18.

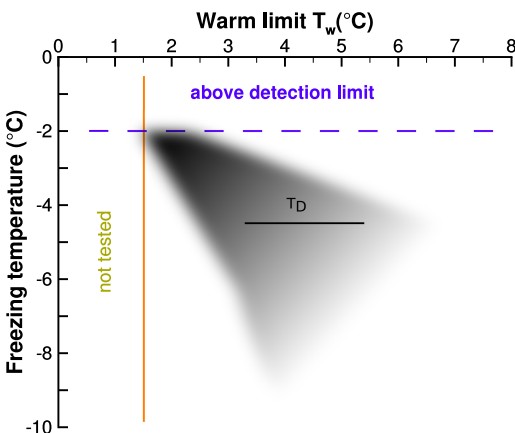

**Figure 18.** Schematic illustration of the boundaries within which PFN were observed. The shading is intended to indicate the likelihood that freezing is observed in various zones.



The experiments of SS01 with aliphatic alcohol layers as ice nucleators can not be directly compared with those for $HGI_2$. However, there are qualitative agreements on all aspects of pre-activation. Their results confirm the existence of a warm temperature limit for PFN to occur: they give the warm limit $T_D = +27.5 \pm 2.5°C$ and for pentacosanol (C25) $T_D = +35 \pm 2.5°C$. The tolerances attached to the values reflect the fact that the loss of PFN is gradual near $T_D$. This is in accord with the findings presented. There is no mention in SS01 of the need to reach $T_C$ and they also show a large spread in $T_f^*$. Results in SS01 show an even stronger difference from the EEZ70 claim for freezing near $0°C$ after pre-activation. For the most active aliphatic alcohol nucleant, SS01 report no freezing events with $T_f^* > -5°C$ and with the mean near $\overline{T_f^*} > -8°C$.

### 5.2.1 Repeatability of PFN

Repeatability of observed freezing temperatures under various conditions is a useful means for considering possible mechanisms responsible for the PFN and to examine the potential roles of nucleation sites in particular.

The correlation coefficients shown in Table 3 provide one measure of stability. The majority of the correlation coefficients are in the range $r = 0.5...0.9$. These values are lower than the values reported in Vali (2008) for soil suspensions and for distilled water. However, randomization tests showed that the $r$ values are significantly larger than would be expected for random mixing of drops.

A more focussed discussion about repeatability relies examinations of the sequences of freezing temperatures of individual drops, for sets of many tens of drops taken from the same sample and exposed to the same temperature histories. As mentioned in Section 3.1 there was no control over the number of $HgI_2$ particles per drop, but that number was almost certainly large enough ($\approx 10^{11}$) to consider any observed freezing event to be unique and the repeatability of that freezing events in subsequent cycles to be indicative of the potential stability, or impermanence, of the INP responsible for the event.

The "spiderweb" diagrams in Figs.9, 11 and 12 are graphic representation of drop histories. For comparison, a sample "spiderweb" is included for a soil sample in Fig. 19. The prevalence of roughly horizontal lines in this diagram is an indication of a degree of repeatability. The median correlation coefficient for pairs taken from these 20 runs is 0.92; the 90% range is $r = 0.6 \rightarrow 0.96$. Sudden changes as well as minor ups and downs in the spiderweb are also evident in this figure. Strictly speaking such changes make the view of sites as permanent untenable, but can also be interpreted as variations occasionally superimposed on permanence. The relative frequency of near-constancy versus variability lies at the heart of the debate in the literature about nucleating sites.

One reasonably firm result from these experiments is that there is no relationship between between the freezing temperature $T_f$ on first freezing or in subsequent cycles with $T_w > T_D$ and $T_f^*$ with pre-activation. This suggests that the pre-activation observed in these experiments is not an enhancement of the ability of a given site to nucleate ice, but that different sites are responsible for the nucleation with or without pre-activation.

In spite of the limited potential of these experiments for examining the repeatability of PFN, Section 4.4 presents evidence for limited repeatability. The evidence is a cumulative one of various different test results. It is reinforced by the rare but not insignificant number of sequences of freezing events that are prime examples of repeatability.





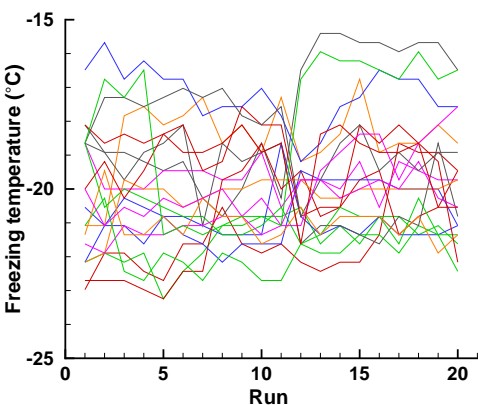

**Figure 19.** "Spiderweb" diagram of freezing temperatures of 20 drops of a soil suspension in 20 runs from a subset of data in Vali (2008).

The above conclusion differs from results shown in SS01 for many repetitions of the freeze cycle for a single drop. They

observed an approximately $4°C$ random spread of $T_f^*$ over 140 repetitions. Roughly the same spread was associated with each $T_w$ and even for $T_w > T_D$. The spread was also about the same for the three different aliphatic alcohols. This would indicate something specific to that nucleant class. For that reason, those results should not be viewed as direct contradictions with the results presented for repeatability.

### 5.3 Explanations of PFN

A summary of the empirical evidence for PFN and the explanations given for it has been given in Section 2. Those explanations, as many others in the literature about pre-activation in various systems centered on two alternatives. Special features on the substrate such a cavities, cracks, etc. were suggested to lead to PFN by retaining some of the solid phase past the bulk melting point (Turnbull, 1950). Alternatively, PFN is considered to arise via the formation of an adsorbed ice-like layer enabling PFN to take place past a greatly reduced energy barrier (e.g. Mason 1950, 1956, Mason and Maybank 1958). Neither explanation

has been confirmed by direct evidence such as detection of the retained ice phase or of the ordered layer past the melting point. Arguments supporting the role of cavities are based on examinations of the thermodynamic conditions for phase stability of confined water or ice. Pore condensation and freezing (PCF) is explained this way (Marcolli, 2020). Support for the role of ice-like layers was found by EEZ70 in the independence of $T_D$ on pressure or on the presence of dissolved salts. SS01 found support for the ice-like layer explanation in the shift of $R(T)$ to lower temperature with increasing chain length of the alcohol

layer without much change in the shape of that function. In essence, both EEZ70 and SS01 rely on criteria linked to the limit of survival of the structure responsible for PFN above the melting point.

This work was not undertaken to test either theory. The explanations offered in EEZ70 and in SS01 for the pre-activation on terms of an adsorbed ice-like layer on the substrate has no direct support in this work. Tests were made only with $HgI_2$, and at only one pressure. But none of the present results contradict the role of an ice-like layer in PFN.





To reconcile repeatability, and hence distinct differences among the INPs involved in producing PFN, with the ice-like layer explanation that theory must be first modified to have some characteristic of the layer differ from one case to the other. This could be occurring via variations in the thickness, size, or completeness of the ice-like layer. For example, it could be assumed that the layer is more like a patch than an extended layer. The size of the patch then can be assumed to be influenced by some underlying feature of the substrate. In this way, the patch becomes an extension of a site. Furthermore, the explanation has

to be confronted with the evidence that there is no relationship between the initial nucleation temperature of a site and the pre-activated nucleation temperature, as shown in Section 5.2.1. In answer to that, it is not difficult to imagine that any ice-like patch would form on different surface features than the ones leading to nucleation without pre-activation. The difference between temperatures $T - f$ and $T_f^*$ is in itself an indication that different sizes of ice-binding locations are involved. The spread in observed upper limits $T_D$ could also be a consequence of different size patches melting at different temperatures.

This is clearly a heuristic solution and is given here as an example of the type of model that will have to be developed to account for the known characteristics of PFN.

   While finding support for the ice-like layer model, SS01 also raise the possibility of pre-activation arising via changes in the Langmuir layer of the aliphatic alcohol. They describe this as "... a metastable conformation of the alcohol layer itself ... the freezing process induces a phase transition of the alcohol film". Furthermore, "... based on the known chain length dependence

of two-dimensional melting temperatures for fatty acid monolayers, it is not unreasonable that the increase which we observe in $T_D$ with chain length could in fact be related to a structural change of the Langmuir film". However, as argued in the preceding paragraph, the modification of the alcohol film would also have to be thought of as having different "patch sizes" or other characteristics differing from drop to drop and having some degree of stability.

   The idea expressed in the foregoing paragraph can perhaps be extended to thinking about PFN on $HgI_2$ too as some modifi-

cation of the substrate surface. The fact that $HgI_2$ is said to be a soft substance helps such thinking. More directly, the weaker correlations found for $HgI_2$ for runs without pre-activation, in comparison with mineral nucleants, also indicate some "softness" or malleability of the substrate on the scale of ice embryos. The large changes observed from one cycle to the next, both with and without pre-activation, further encourage thinking about changes on the substrate itself between cycles.

   In reality, the line is blurred between considering surface sites, or ice-like patches of specific character, to explain PFN.

SS01 also came to the conclusion that a "... complete explanation of the pre-activation mechanism will treat the monolayer and the vecinal water as a strongly interacting system." (Monolayer in this sentence refers to the alcohol covering of the drop that serves as the nucleant.) The main point raised by the results presented here is that the PFN mechanism is site dependent in a fashion similar to the way it is for nucleation in general.

   It is also important to consider the reverse of repeatability, namely the degree of variability observed in much of the data.

Molecular fluctuation of water molecules associated with embryo formation are an inescapable part of nucleation. Additional factors may be the fragility of the structures making up the sites. For PFN, it is clear that the structure forms after previous formation of ice on the nucleator surface. Whether this is an imprint on that surface, or retention of an adsorbed patch of water molecules ordered into ice-like configuration, the structure is likely to be less rigid than what are normally considered potential sites. The fragility of PFN structures on $HgI_2$ is perhaps pre-conditioned by the relatively weak permanence of sites





on the $HgI_2$ surface in comparison with other materials. The low values of the correlation coefficients given in Table 3 for runs without PFN underscores this possibility.

## 6    Conclusions

As already indicated in the title of this paper, the work described here has instrumental and procedural limitations and hence the results have to be viewed as exploratory. The conclusions given below are in part based on the previous work of EEZ70

and SS01, and in part on differences from those.

1. Pre-activated ice nucleation, PFN, on $HgI_2$, as reported in EEZ70 and as schematically depicted in Fig.1, was confirmed. PFN leads to freezing at temperatures much higher than would otherwise occur with the same substance.

2. PFN was observed at $T_f^* \geq -2°C$ but no determination was made of a minimum supercooling for PFN. PFN is exhibited over a temperature range extending at least to $-6°C$.

3. The upper limit of warming above $0°C$ that preserves PFN was found to be a transition not a fixed value. Raising the degree of heating leads, at first to a gradual lowering of the subsequent nucleation temperature $T_f^*$, then to a large drop at $T_D \approx +3.5°C$, and then extending with PFN still clearly evident to as high as $T_D \approx +5°C$. Beyond this overall trend, there is a considerable variation in the manifestation of this pattern for individual INPs contained in separate drops.

4. There is no support in these experiments for the existence of a temperature limit to which a sample has to be cooled for

PFN to be exhibited. Freezing and limited warming are sufficient conditions.

5. No relationship was found between the initial nucleation temperature $T_f$ of a given drop and its nucleation temperature $T_f^*$ after pre-activation. This raises the possibility that different particles or different parts of particle surfaces are involved in the two cases.

6. Repeatability, as defined at the beginning of Section 4, is supported to a limited degree but the evidence may be con-

sidered unequivocal when manifested. On the other hand, much of the data points to a lack of repeatability of PFN, indicating the fragility of the features leading to the partial repeatability observed.

7. The evidence for the possibility of repeatability emphasizes the role of specific features, sites, leading to PFN just as it has in nucleation in general.

8. $HgI_2$ is a relatively poor source of INPs and some unexplained variations were found in INP concentration per unit mass

during the course of these experiments. These variations had negligible impact on the results regarding PFN but may be related to the lower degree of stability of sites than observed for some other materials.

The results presented here will, hopefully be re-examined in future tests. The two conflicting aspects - the evidence for sites and the paucity and fragility of the sites - are challenging aspects to study. The range of warm limit conditions which lead to



only partial PFN (not all drops freeze at the same temperature) may be a specially fruitful situation to study. Experiments with
other materials exhibiting PFN would be useful.

## 6.1 Broader context

The abundance of mercuric iodide in the atmosphere is not known. Atmospheric concnetrations of mercury and of its com-
pounds and hte reactions connecting these, as well as the surface sources and removal processes have been subjects of study for
decades. Schroeder and Munthe (1998) and Lyman et al. (2020) present overviews of the physical, chemical and toxicological
properties of mercury in vaarious forms. No mention is made of mercuric iodide in these papers indicating that it has no known
importance in atmospheric processes. The analyses of atmospheric ice nucleating particles (INPs) have not revealed mercuric
iodide as a component.

In addition to mercuridc iodide, PFN was observed by EEZ for lead iodide, gypsum, cadmium iodide, muscovite, L-
asparagine, L-aspartic acid and p-benzyl phenol. The absence of PFN was reported for graphitem chlorite, silica gel and a
number of other inorganic materials. Evans (1967) reported PFN on phloglucinol dihydrate. These findings were not stated to
be the result of exhaustive search for materials that would exhibit PFN, so it is reasonable to assume that further research will
reveal PFN associated with yet more substances. The detailed character of PFN for the substances listed above and for others
yet to be identified will have to be determined in future research. There is a reasonable probability that the general features of
PFN identified for mercuric iodide will also hold for most other substancs. The basis for this assertion is that the findings are
in concert with other evidence for ice nucleation in the immersion mode.

Since pre-ectivation is a sequence of exposures to different temperatures, atmospheric impact of the process will also depend
on the occurence of such sequences in clouds. Vertical circulation of parcels in clouds is well known to occur in deep convection
and in extended layer clouds. For mercuric iodide the circulation would have to extend over a kilometer or more, which is
possible but not frequent. In any event, freezing at temperatures slightly below the melting point is of great importance for
clouds and for many biological and environmental systems, as well as for artificail snow making, tissue preservation and more.
Thus, PFN may have to be considered as a porrible pathway for natural and induced freezing at minimal supercooling.

From the point of view of nucleation basics, the results presented in this paper underscore the need for more fundamental
understanding of what surface structures constitues nucleation sites, how stable they are, and what infuleces are excerted
by time, temperature and other factors. In that sense, PFN is an aspect of ice nucleation that presents further options for
characterizing the process.

*Data availability.* Raw data can be made available by the author. It will be placed in a repository before final publication.

*Author contributions.* All the work reported here was done by the author.



**Table 5.** Nomenclature.

| | |
|---|---|
| CNT | Classical nucleation theory |
| i subscript | attached to $T_w$ or $T_f$ refers to specific run number or specific drop number, depending on context |
| INP | Ice nucleating particle, as defined in Vali et al. (2015) |
| PFN | Pre-activated freezing nucleation |
| $\omega$ | equation (7) in Vali (2014): $\omega = -d(lnR)/dT$ |
| $r$ | correlation coefficient |
| $R(T)$ | freezing rate in SS01 and Vali (2014) |
| $T_f$ | observed temperature of freezing of a drop, or as a general reference to nucleation temperature |
| $T_f^*$ | observed freezing temperature of a drop with pre-activation |
| $T_w$ | temperature to which the sample is heated between runs; the "warm limit". |
| $T_C$ | defined in EEZ70 as the temperature to which a sample has to be cooled for PFN to be manifested |
| $T_D$ | limiting value for $T_w$ for retaining PFN |

*Competing interests.* The author declares that there is no cempeting interest.

*Acknowledgements.* Financial support for the work was provided by the U.S. Bureau of Reclamation via contracts with the University of
Wyoming. Thanks are due to Dr. Donald L. Veal (†) for supporting this research in many ways. The contributions of laboratory assistants in
carrying out the experiments and in data reduction is much appreciated.

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
