# Peer review of "EXPLORATORY EXPERIMENTS ON PRE-ACTIVATED FREEZING NUCLEATION ON MERCURIC IODIDE"

_Atmospheric Chemistry and Physics, 2020_

## Referee Comment (RC1) · Thomas Leisner (Referee) · 26 Oct 2020

The manuscript discusses experimental results on the phenomenon of pre-activated freezing nucleation (PFN), which was then a topic of discussion, but has received renewed interest recently. In contrast to pore condensation it is a phenomenon associated with immersion freezing and is operational at temperatures only slightly below the melting point. It is therefore of great interest to mixed phase cloud research.

The submission is somewhat unusual as it describes experiments which were performed 48 years ago in the laboratory of the author but have not been published comprehensively since then. The experiments deal with HgI_2 as an nucleant and constitute quantitatively an impressive piece of work, especially when the experimental

means of the 1970′s are taken into account. The methods and the results are described comprehensively and the effect of PFN is clearly worked out. In particular, the range of the warm limit temperature, up to which the effect is sustained, has been thoroughly characterized. The author shows convincingly that "active sites" responsible for PFN are not identical to the sites responsible for heterogeneous freezing without PFN. A large part of the manuscript discusses variability on a single droplet level and repeatability between subsequent cooling cycles. It is surprising that the droplet- to droplet variability is high in parallel cooling cycles even though each droplet contains about $10^{11}$ particles of HgI_2 according to the authors estimate. This touches on one (not mendable) weakness of the manuscript, which is the fact that the samples are no longer available and neither particle size distribution nor particle concentration were measured back then. The manuscript carefully mentions and discusses this and other deficiencies of the experiment and their implications for the interpretation of the results.

The author puts his results into a stringent conceptual background and discusses the implications for a mechanistic understanding of heterogeneous ice formation and possible ramifications for cloud research.

I clearly recommend to publish this manuscript once my remark below is addressed. I hope that the publication will advance more work on this fascinating effect and will foster the search for more materials, possibly of greater atmospheric relevance, that exhibit PFN.

Remark: I am not convinced that the more qualitative figure 18 is helpful in its current form. In particular, I cannot easily see how it is quantitatively in accord with the data presented earlier, e.g. in Figure 3, which to my understanding, should map out the same space if the vertical bars are taken into account. I would suggest to either remove figure 18 or to augment it with overlaid experimental data, e.g. in the form of a box-whisker plot.

Minor remark: There are several typographical errors that should be corrected, e.g.

[Figure]

"understading" in the abstract, typos are particularly frequent in paragraph 6.1

---

## Referee Comment (RC2) · Anonymous Referee #2 · 17 Nov 2020

Review of "Exploratory experiments on pre-activated freezing nucleation on mercuric iodide" by Vali.

In this manuscript, Vali presents results from experiments carried out in the 1970s on the pre-activated freezing nucleation caused by mercuric iodide. A very thorough analysis of the data is presented, and the results are certainly interesting. Although mercuric iodide may not be atmospherically relevant, the results may be applicable to other substances of atmospheric relevance, as pointed out by the author. I support publication in Atmospheric Chemistry and Physics, after the author has adequately addressed the following minor comments:

1. The abstract is rather vague. The author may want to include some of the specific conclusions from Section 6.

[Figure]

2. There are several typos throughout the document, especially Section 6.1. For example, see the following: Page 13, line 228; Table 4 Caption; Page 21, line 343, 349, 359; Page 23, line 376; Page 25, line 438; Page 27, line 498.

3. Figure 2. The figure caption states "cumulative nucleus spectra, K(T)", but the y-axis states "differential concentrations, k(T)". Are these the same thing? I assume cumulative is different from differential? I also suggest adding the equation that describes the cumulative concentration of INP to the manuscript, rather than refer to the equation in Vali et al. 2014. This way a reader will not have to look up another paper to fully understand the current paper.

4. Table 1. please indicate the units displayed in the table.

5. Table 2. This table took some time to understand. I suggest labeling the top row and include units. Also, in the figure caption, please indicate what "cutoff" represents.

6. Page 9, lines 184-185. "these differences are statistically significant to better than 001 % level." What statistical test was used?

7. Results for Exp. I were presented (e.g. Table 3) before describing the temperature histories used in these experiments. I found this confusing.

8. Page 21, line 331. The symbol omega is used but only defined in Table 5. Should the symbol also be defined in the main text? Also, Table 5 is not referred to in the main text. Hence, the reader may not automatically find the definition for omega.

9. In Section 5.3, the results of this paper are discussed in terms of an adsorbed ice-like layer on the substrate. The author states "but none of the present results contradict the role of an ice-like layer in the PFN". Do the results contradict the role of special features on substrates such as cavities, cracks, etc. in the PFN? I.e. do the result contradict the suggested mechanism by Turnbull [1950]? This was not clear to me.

10. Figure 18. This figure does not appear to be quantitatively consistent with the results presented in the current manuscript. The authors should adjust to make the

figure more quantitatively consistent with the data or consider removing the figure.

11. Since the paper has 19 figures and 5 tables, I wondered if some of the figures and tables could be moved to the supplement to make the paper more succinct.

---

## Author Comment (AC1) · 13 Jan 2021

**Replies to comments by referee #1:**

January 12, 2021

Original comments in italics; reply in roman letters. The revised manuscript is appended.

Many thanks for the support expressed for publication of this paper in spite of the unusually long lapse of time since the experiments were performed. Since the topic has not been taken up over the past decades, these results are still novel contributions.

I clearly recommend to publish this manuscript once my remark below is addressed. I hope that the publication will advance more work on this fascinating effect and will foster the search for more materials, possibly of greater atmospheric relevance, that exhibit PFN.

It isn't unrealistic to think that other materials will also exhibit PFN since the materials already known to have that property are quite varied. Looking for PFN on INPs detected in precipitation, or in atmospheric aerosol in general may lead to important findings. Also, developing cloud seeding materials that have been pre-activated seems like a promising possibility..

Remark: I am not convinced that the more qualitative figure 18 is helpful in its current form. In particular, I cannot easily see how it is quantitatively in accord with the data presented earlier, e.g. in Figure 3, which to my understanding, should map out the same space if the vertical bars are taken into account. I would suggest to either remove figure 18 or to augment it with overlaid experimental data, e.g. in the form of a box- whisker plot.

My intention with this figure was to give a quick, suggestive summary of the results and to contrast them with the concept of a well-deined  $T_D$  indicated by earlier research. Variations among the various experiments, due to different particle concentrations and other factors, made Fig. 18 qualitative rather than quantitatively precise. Since both reviewers found this diagram problematic, the figure will be removed from the final version.

1

Thanks for pointing to the typos. Corrected.

---

## Author Comment (AC2) · 13 Jan 2021

**Replies to comments by referee #2:**

January 12, 2021

Original comments in italics; reply in roman letters. Page and line references in the original comments refer to the original manuscript. In the responses, page references to the changes made are in brackets [...].

The revised manuscript is appended.

Many thanks for the suggested corrections and clarifications; all were helpful. I hope that the revised manuscript has fewer flaws and errors.

1. The abstract is rather vague. The author may want to include some of the specific conclusions from Section 6.

R: Thanks for pointing this out. Made change in the Abstract to correct. [2 ... 18]

2. There are several typos throughout the document, especially Section 6.1. For ex- ample, see the following: Page 13, line 228; Table 4 Caption; Page 21, line 343, 349, 359; Page 23, line 376; Page 25, line 438; Page 27, line 498.

R: Corrections made.

3. Figure 2. The figure caption states "cumulative nucleus spectra, K(T)", but the y-axis states "differential concentrations, k(T)". Are these the same thing? I assume cumulative is different from differential? I also suggest adding the equation that describes the cumulative concentration of INP to the manuscript, rather than refer to the equation in Vali et al. 2014. This way a reader will not have to look up another paper to fully understand the current paper.

R: The figure caption was incorrect. The equation for k(T) was added to the text and the new symbols were added to Table 5. [155 ... 158]

4. Table 1. please indicate the units displayed in the table.

R: All temperatures are in °C. Indicated this in the table heading and also in Table 5. [187]

5. Table 2. This table took some time to understand. I suggest labeling the top row and include units. Also, in the figure caption, please indicate what "cutoff" represents.

R: The table and the caption have been revised. [page 9]

6. Page 9, lines 184-185. "these differences are statistically significant to better than 001 % level." What statistical test was used?

R: The result quoted is from the difference of means test. Added clarification and reference in the manuscript. [204]

7. Results for Exp. I were presented (e.g. Table 3) before describing the temperature histories used in these experiments. I found this confusing.

R: Included a reference where the experiment is first mention to the section where Exp I is detailed. [226]

8. Page 21, line 331. The symbol omega is used but only defined in Table 5. Should the symbol also be defined in the main text? Also, Table 5 is not referred to in the main text. Hence, the reader may not automatically find the definition for omega.

R: The sentence where  $\omega$  is mentions defines it as "the slope of the logarithm of concentration". Didn't want to burden the paragraph by including the equation for  $\omega$  there, but it is included in Table 5. [343... 344]

9. In Section 5.3, the results of this paper are discussed in terms of an adsorbed ice-like layer on the substrate. The author states "but none of the present results contradict the role of an ice-like layer in the PFN". Do the results contradict the role of special features on substrates such as cavities, cracks, etc. in the PFN? I.e. do the result contradict the suggested mechanism by Turnbull (1950)? This was not clear to me.

R: Turnbull's suggestion that special surface features serve as nucleation sites is the general starting point for thinking about nucleation sites. The nature of sites is discussed in the text in more detailed fashion than Turnbull's treatment, so his suggestion is clearly supported and refined. [438...440; 474...475, 479...480; 543...544]

10. Figure 18. This figure does not appear to be quantitatively consistent with the results presented in the current manuscript. The authors should adjust to make the figure more quantitatively consistent with the data or consider removing the figure.

R: My intention with this figure was to give a quick, suggestive summary of the results and to contrast them with the concept of a well-defined  $T_D$  indicated by earlier research. Variations among the various experiments, due to different particle concentrations and other factors, made Fig. 18 qualitative rather than quantitatively precise. Since both reviewers found this diagram problematic, the figure was removed.

**11. Since the paper has 19 figures and 5 tables, I wondered if some of the figures and tables could be moved to the supplement to make the paper more succinct.**

R: I recognize that the paper is indeed fairly long and demanding to read. However, it seems to me important to present to the reader all the complexities of the results. There isn't a basic data set and a simple conclusion to form the core of the paper and to leave the rest to an appendix. Full appreciation of the phenomenon requires the many different aspects of the data to be seen. I feel that breaking out part of the material into an Appendix would make it harder, not easier to see the full picture.

 $\mathbf{2}$